# Population-based assessment of cardiovascular complications of rheumatic heart disease in Fiji: a record-linkage analysis

Tom Parks ,[1,2] Litia Narube,[3] Mai Ling Perman ,[4] Kelera Sakumeni,[5] James J Fong ,[5] Daniel Engelman ,[6] Samantha M Colquhoun ,[7] Andrew C Steer,[6] Joseph Kado [5,8]

For numbered affiliations see end of article.

**Correspondence to**
Dr Tom Parks;
t.parks@imperial.ac.uk

## ABSTRACT

**Objective** To determine population-based rates of non-fatal complications of rheumatic heart disease (RHD).

**Design** Retrospective cohort study based on multiple sources of routine clinical and administrative data amalgamated by probabilistic record-linkage.

**Setting** Fiji, an upper-middle-income country, where most of the population has access to government-funded healthcare services.

**Participants** National cohort of 2116 patients with clinically apparent RHD aged 5–69 years during 2008 and 2012.

**Primary and secondary outcome measures** The primary outcome was hospitalisation for any of heart failure, atrial fibrillation, ischaemic stroke and infective endocarditis. Secondary outcomes were first hospitalisation for each of the complications individually in the national cohort as well as in hospital (n=1300) and maternity (n=210) subsets. Information on outcomes was obtained from discharge diagnoses coded in the hospital patient information system. Population-based rates were obtained using relative survival methods with census data as the denominator.

**Results** Among 2116 patients in the national cohort (median age, 23.3 years; 57.7% women), 546 (25.8%) were hospitalised for an RHD complication, a substantial proportion of all cardiovascular admissions in the country during this period in those aged 0–40 years (heart failure, 210/454, 46.3%; ischaemic stroke 31/134, 23.1%). Absolute numbers of RHD complications peaked during the third decade of life with higher population-based rates in women compared with men (incidence rate ratio 1.4, 95% CI 1.3 to 1.6, p<0.001). Hospitalisation for any RHD complication was associated with substantially increased risk of death (HR 5.4, 95% CI 3.4 to 8.8, p<0.001), especially after the onset of heart failure (HR 6.6, 95% CI 4.8 to 9.1, p<0.001).

**Conclusions** Our study defines the burden of RHD-attributable morbidity in the general population of Fiji, potentially reflecting the situation in low-income and middle-income countries worldwide. Hospitalisation for an RHD complication is associated with markedly increased risk of death, re-emphasising the importance of effective early prevention.

## STRENGTHS AND LIMITATIONS OF THIS STUDY

⇒ Our study was based on probabilistic record-linkage providing an efficient means to derive population-based estimates of disease burden from routinely collected data.

⇒ We used relative survival methods to estimate rates of rheumatic heart disease (RHD)-attributable morbidity in the general populations since this approach helps separate out events that are unrelated to the disease of interest.

⇒ We demonstrate the comparability of the hospital subset of our cohort to key previous hospital-based studies, including the landmark Global Rheumatic Heart Disease Registry (REMEDY) study, set in Asia and Africa, greatly increasing the generalisability of our population-based estimates beyond Fiji.

⇒ The routinely collected data used in our study are likely to contain some inaccuracies, including in the precision of coding of discharge diagnoses. However, there is little reason to suspect that the relevant diagnostic codes would have been used differently in patients with and without RHD.

⇒ Reflecting external factors, we recognise that several years have now passed since the period from which our multiple sources of routine data were available. Nonetheless, they continue to provide invaluable insight into the burden of RHD in low-resource settings, chiefly due to the scarcity of comparable estimates from elsewhere.

## INTRODUCTION

Rheumatic heart disease (RHD) is a chronic disease of the heart valves that predominantly affects children and young adults in low-income and middle-income countries (LMICs).[1] The disease is a consequence of an aberrant immune response to *Streptococcus pyogenes* infection causing inflammation and scarring of the heart valves,[2] which manifests clinically through complications such as heart failure, ischaemic stroke and early death.[3] Secondary prevention is achievable using

long-term antibiotic prophylaxis,[4] although implementation remains challenging in most settings where the disease is endemic.[5] Moreover, there remains no vaccine against the causative *S. pyogenes*.[6]

In recent years, there has been growing recognition that RHD causes a substantial burden of disease and disability globally,[7–10] culminating in 2018 in a World Health Assembly resolution calling for a coordinated global response.[11] Global summary estimates by the Global Burden of Disease (GBD) project suggest RHD accounts for approximately 305 000 deaths and 10.7 million disability-adjusted life years annually, the majority of these occurring LMICs.[7 9 12] Notwithstanding their importance in providing impetus for increased investment in RHD prevention, such estimates remain problematic because of the scarcity of good quality data available from LMIC settings, where the disease is endemic.[13] For example, to our knowledge, our previous record-linkage analysis set in Fiji remains the only population-based estimate of RHD-attributable mortality from an LMIC.[14] Moreover, data pertaining to the rates of non-fatal complications, such as heart failure and stroke, among patients with RHD living in LMICs are particularly scarce.[7]

One compelling exception to this pattern was the landmark Global Rheumatic Heart Disease Registry (REMEDY) study, published in 2016, a prospective, hospital-based registry that defined rates of death, heart failure and stroke among 3343 patients with symptomatic RHD enrolled across 12 African countries, India and Yemen.[15] However, as the authors of that study noted, since the patients were mostly enrolled at tertiary referral hospitals, the rates observed cannot be easily generalised to the wider pool of patients with RHD away from hospital settings.[15] Furthermore, neither REMEDY nor more recent studies of RHD progression provide estimates of population-based rates of complications of RHD,[16–19] which are needed to improve current global summary estimates as well as to inform the design, implementation and evaluation of disease control efforts.[13 20]

Therefore we sought to assess the burden and calculate nationwide population-based rates of RHD-attributable morbidity in Fiji, a middle-income country in the Western Pacific.

## METHODS
This study investigated RHD-attributable morbidity among patients with clinically apparent RHD in Fiji during 2008–2012. Patients were identified from a cohort previously ascertained by probabilistic record linkage (online supplemental figure S1).[14] In this analysis, we investigated the four major complications of RHD: heart failure, atrial fibrillation, ischaemic stroke and infective endocarditis,[3] which together we term 'any complication'. We then used census data to calculate age-specific RHD-attributable rates, as well as estimates of years lived with disability (YLDs), for the wider population.

We also examined two subsets of the 'national' cohort in further detail (online supplemental figure S1). The first, termed the 'hospital' cohort, comprised patients with at least one hospital attendance during follow-up, allowing assessment of the relationship between RHD-attributable morbidity and mortality in this setting, as well as comparison with previously reported hospital-based studies. The second, termed the 'maternity' cohort, comprised women with at least one diagnostic code indicating pregnancy or childbirth during follow-up, enabling a description of the RHD-attributable morbidity and mortality during pregnancy, childbirth and up to 1-year postpartum in the cohort.[21]

### Setting
At the time of the study, the population of Fiji was approximately 840 000, comprising two major ethnic groups, Indigenous iTaukei and Fijians of Indian Descent.[22 23] Government-funded hospital services are available free-of-charge including consultations, admissions, laboratory investigations and radiological examinations.[24] A high proportion of pregnant women give birth in hospital.[24]

### Cohort
We used sources of data and record-linkage methods as described previously.[14] Briefly, we used a probabilistic record-linkage procedure to amalgamate data across the four data sets: a hospital patient information system, a disease control register, echocardiography clinic registers and a database of death certificates. After extensive cleaning and standardisation, we used multiple identifiers including registration numbers, names, sex, ethnicity, birth dates and locality of residence to find pairs of records that referred to the same individual. In its final configuration, our procedure identified known duplications in the hospital patient information system with sensitivity of 91.4% and specificity of 99.9%.[14]

For this current analysis, we focused on patients with clinically-apparent RHD reported in one or more of the hospital patient information system, the disease control register and the echocardiography clinic registers. As before, we restricted the analysis to patients aged 5–69 years and the period 2008–2012 for which the most data was available.[14] We obtained information on complications resulting in hospitalisation, as well as pregnancy outcomes from the coded discharge diagnoses reported in the hospital patient information system. Data were available from all government-funded hospitals, including the two regional and one national referral hospitals, but not from the smaller private sector[24] The completeness and quality of Fiji's hospital patient information system data have been independently investigated previously[25] and the diagnostic codes used to define RHD and its complications are summarised in online supplemental table S1. We also used this system to obtain information on comorbidities using a system based on the Charlson Comorbidity Index[26] but excluding heart failure and

stroke from the calculation since these were outcomes of interest in this study.

The primary outcome was first hospitalisation for any complication defined by the date of hospital admission when the relevant diagnostic code first appeared in the patient's records. The secondary outcomes were hospitalisation for each of the complications individually. We assumed no loss to follow-up because we expected the patient information system would include most hospitalisations in the country during 2008–2012.[24 25]

## Statistical methods

We used survival methods to calculate cumulative incidence rates for each outcome in the cohort. Notably, the available coded discharge diagnoses data did not reliably distinguish between new and past diagnoses. Accordingly, to minimise the risk of counting events that occurred prior to the study, we limited the incidence rate calculations for the three more frequent outcomes, any complication, heart failure and atrial fibrillation, to the fourth and fifth year of the study. However, for the two remaining outcomes, ischaemic stroke and infective endocarditis, we used all the data available because: (1) these less frequent outcomes occurred at a comparatively constant rate throughout the study (online supplemental figure S2); and (2) preliminary estimates based on the fourth and fifth year of the study tended to be imprecise. Having excluded nine individuals with missing sex information, we then used competing-risks regression (Stata *stcrreg* command)[27] to examine the impact of age (in three categories), sex and ethnicity (both as recorded in the hospital patient information system) and comorbidity on the risk of complications with death as the competing failure event.

To obtain rates of RHD-attributable morbidity in the general population, we used relative survival methods by applying background rates of hospitalisation to the cohort by sex, ethnicity and age in thirteen 5-year age categories.[28] Background rates for each diagnosis were estimated by Poisson regression using the counts of hospitalisations from the patient information system and concurrent denominator population estimates published by the Fiji Bureau of Statistics.[22] Specifically, we calculated age-specific rates for each complication using the observed minus the expected number (ie, the excess) as the numerator divided by the population estimates as the denominator with 95% CIs calculated using Poisson.

Years lived with disability were calculated by applying disability weights used by the GBD study to the person-time of the cohort during follow-up using weights: heart failure (severe), 0.179; stroke (moderate), 0.316; atrial fibrillation (symptomatic), 0.224; infective endocarditis (severe), 0.133; uncomplicated RHD, 0.049.[12] Where a patient had experienced more than one complication, disability weights were combined using a multiplicative model, as previously described.[29] To estimate disability-adjusted life-years (DALYs), we updated our previous estimates of years of life lost (YLLs)[14] using the GBD 2019 theoretical minimum risk life table.[12] In addition to summary estimates for the population of Fiji aged 0–69 years, we also calculated age-standardised rates using the GBD world population age standard[30] with rates of death and disability from our study for the population aged 65–69 years applied to the population aged≥70 years.

To assess the relationship between RHD-attributable morbidity and mortality in the cohort, we focused on the hospital cohort subset, defined by at least one attendance during follow-up at one of the two regional hospitals or the one national referral hospitals in the country. Alongside the opportunity for comparison with previously reported hospital-based studies, this subgroup had the advantage of a well-defined starting point for the time-to-event analysis (ie, the earliest recorded hospital or outpatient clinic attendance in the study period at which a diagnosis of RHD was recorded) as well as the availability of additional clinical parameters for analysis (eg, disease severity and presence of mitral stenosis). For each complication, having excluded five individuals with missing sex information, we split the follow-up time at the earliest point at which the complication occurred (Stata *stsplit* command) and plotted Kaplan-Meier estimates of survival based on the resulting variable to illustrate risk before and after the event occurred. We then used a Cox regression model to measure the relative impact of each of the complications, using the Schoenfeld residuals to test the proportional-hazards assumption (Stata *phtest* command). All models included potential confounders—age, sex, ethnicity, comorbidity and rural residence—and evaluated interactions using likelihood ratio tests. Statistical analyses were conducted using Stata, V.12 (StataCorp, College Station, Texas, USA).

## Patient and public involvement

Patients and/or the public were not involved in the design, or conduct, or reporting, or dissemination plans of this research.

## RESULTS

From our previously reported cohort of 2619 patients,[14] we excluded 92 individuals known only for acute rheumatic fever, 302 identified only through echocardiographic screening and 109 for whom the only available information was a death certificate (online supplemental figure S1). The cohort comprised 2116 patients with clinically-apparent RHD followed for a median of 5.0 years. Most were aged 5–39 years (74.6%) and more than half were women (57.7%, table 1).

## RHD-attributable morbidity in the cohort

There were 2116 patients in the national cohort, totalling 9439 years of person-time. During follow-up, 1335 (63.1%) patients were hospitalised a total of 2896 times, equating to 1.4% of the total 205 427 admissions for any cause in the population of Fiji aged 5–69 years during 2008–2012. The most frequent discharge diagnostic

**Table 1** Baseline characteristics of patients by sex in the national, hospital and maternity cohorts*

| | National (n=2116) | | Hospital (n=1300) | | Maternity (n=210) |
|---|---|---|---|---|---|
| | Male (n=892) | Female (n=1215) | Male (n=547) | Female (n=748) | Female (n=210) |
| **Age, n (%)** | | | | | |
| 5–14 years | 356 (39.9) | 326 (26.8) | 197 (36.0) | 179 (23.9) | 5 (0.95) |
| 15–39 years | 312 (35.0) | 577 (47.5) | 191 (34.9) | 356 (47.6) | 202 (96.2) |
| 40–69 years | 224 (25.1) | 312 (25.7) | 159 (29.1) | 213 (28.5) | 6 (2.9) |
| **Ethnicity, n (%)** | | | | | |
| iTaukei (Indigenous) | 527 (59.1) | 775 (63.8) | 332 (60.7) | 500 (66.8) | 151 (72.3) |
| Fijian of Indian Descent | 294 (33.0) | 366 (30.1) | 166 (30.4) | 201 (26.9) | 41 (19.6) |
| Other | 71 (8.0) | 74 (6.1) | 49 (9.0) | 47 (6.3) | 17 (8.1) |
| **Data sources, n (%)** | | | | | |
| Hospital admission | 375 (42.0) | 562 (46.3) | 324 (59.2) | 461 (61.6) | 135 (64.3) |
| Echocardiography clinic | 427 (47.9) | 608 (50.0) | 341 (62.3) | 480 (64.2) | 129 (61.4) |
| Control programme | 523 (58.6) | 682 (56.1) | 185 (33.8) | 223 (29.8) | 92 (43.8) |
| **Comorbidity†, n (%)** | | | | | |
| Index=0 | 763 (85.5) | 1047 (86.2) | 435 (79.5) | 615 (82.2) | 195 (92.9) |
| Index=1 | 48 (5.4) | 46 (3.8) | 41 (7.5) | 33 (4.4) | 7 (3.3) |
| Index≥2 | 81 (9.1) | 122 (10.0) | 71 (13.0) | 100 (13.4) | 8 (3.8) |
| Rural residence, n (%) | 400 (44.8) | 515 (42.4) | 245 (44.8) | 301 (40.2) | 64 (30.5) |
| Pregnancy, n (%) | 0 | 210 (17.3) | 0 | 155 (20.7) | 210 (100) |

*Sex information missing for nine patients in the national cohort and five patients in the hospital cohort.
†Modified Charlson Comorbidity[26] calculated without points awarded for heart failure and stroke.

codes were heart failure (22.5% of admissions) and atrial fibrillation (17.6%) (online supplemental table S2).

In total, 546 patients (25.8%) were admitted with a discharge diagnosis consistent with a complication of RHD (figure 1). More specifically, 349 patients (16.5%) were admitted with heart failure, 268 (12.7%) for atrial fibrillation, 104 (4.9%) for ischaemic stroke and 73 (3.4%) for endocarditis (online supplemental table S3). Furthermore,

220 (10.4%) patients had admissions for more than one complication of which heart failure and atrial fibrillation in 154 (7.3%) followed by ischaemic stroke and atrial fibrillation in 46 (2.2%) were the most frequent combinations. The yearly cumulative incidence rate of any complication was 4.6% per year (95% CI 3.9 to 5.4) with rates of all complications except infective endocarditis increasing with age (table 2 and online supplemental table S4).

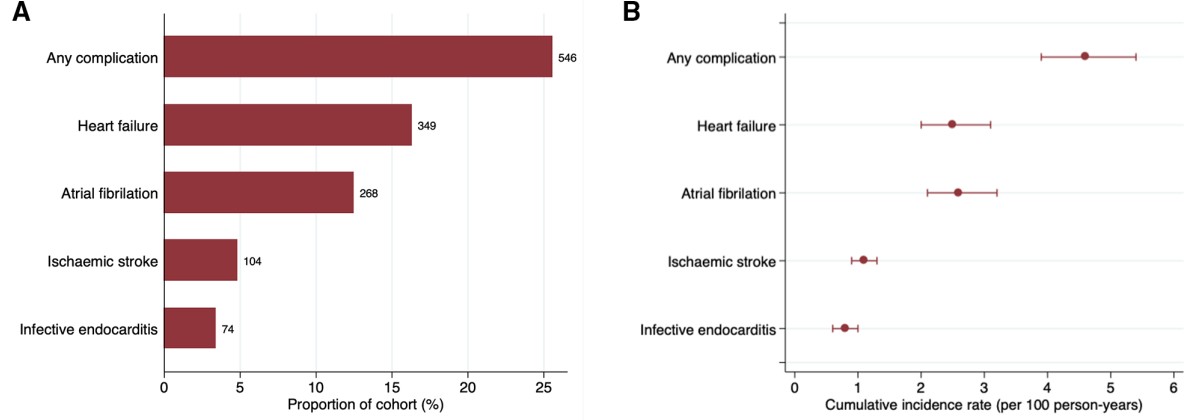

**Figure 1** Hospitalisations for non-fatal complications in the cohort. (A) Proportion of national cohorts hospitalised for each complication of rheumatic heart disease with the number of reaching each outcome indicated at the end of each bar. (B) Cumulative incidence rates of each complication with 95% CIs.

**Table 2** Cumulative incidence rates per 100 person-years with 95% CIs of non-fatal complications of rheumatic heart disease within the national cohort*

| Age | Sex | Any complication† | Heart failure | Atrial fibrillation | Ischaemic stroke | Infective endocarditis |
|---|---|---|---|---|---|---|
| 5–14 years | Male | 1.7 (0.8 to 3.6) | 1.2 (0.5 to 2.9) | 0 | 0.2 (0.0 to 0.7) | 0.9 (0.5 to 1.6) |
| | Female | 1.1 (0.4 to 2.9) | 0.8 (0.3 to 2.5) | 0 | 0.2 (0.0 to 0.7) | 0.5 (0.2 to 1.2) |
| 15–39 years | Male | 4.9 (3.5 to 7.0) | 2.7 (1.7 to 4.2) | 2.1 (1.2 to 3.5) | 1.0 (0.6 to 1.6) | 1.4 (0.9 to 2.1) |
| | Female | 2.8 (2.0 to 4.0) | 1.5 (0.9 to 2.4) | 1.5 (0.9 to 2.4) | 0.7 (0.4 to 1.0) | 0.5 (0.3 to 0.9) |
| 40–69 years | Male | 12.8 (9.9 to 18.3) | 6.5 (4.2 to 10.3) | 6.6 (4.3 to 10.2) | 2.9 (2.0 to 4.2) | 1.0 (0.5 to 1.9) |
| | Female | 10.3 (7.5 to 14.2) | 4.8 (3.2 to 7.4) | 8.5 (6.2 to 11.8) | 2.6 (1.8 to 3.6) | 0.4 (0.2 to 0.9) |

*Rates are based on hospitalisation during the fourth or fifth year for any complications, heart failure and atrial fibrillation, but otherwise at any stage during the study. Rates for the entire study period are given in online supplemental table S4.
†At least one hospitalisation for heart failure, stroke, atrial fibrillation or endocarditis.

In a multivariate analysis with adjustment for age and comorbidity, complications occurred more frequently in men than in women (HR 1.6, 95% CI 1.2 to 2.3, p=0.005, online supplemental table S5). In addition, Indigenous iTaukei or other ethnicity was associated with a more than twofold increased risk of complications compared with Fijians of Indian Descent (HR, 2.7, 95% CI 1.8 to 4.0, p<0.001). Across the complications, residential location (urban/rural) had no impact on risk (p>0.20).

### RHD-attributable morbidity in the general population

To estimate RHD-attributable morbidity in the general population, we compared rates of hospitalisation for complications to those in the general population. Among the population of 5–69 years in Fiji during 2008–2012, 651 of 4756 (13.7%) hospitalisations with a discharge diagnosis consistent with heart failure, 509 of 2225 (22.9%) with atrial fibrillation, 127 of 1927 (6.6%) with ischaemic stroke and 73 of 108 (67.6%) with endocarditis occurred in patients with an underlying diagnosis of RHD (figure 2). However, before 40 years of age, 210 of 454 (46.3%) hospitalisations with heart failure, 143 of 288 (49.7%) with atrial fibrillation and 31 of 134 (23.1%) for stroke were associated with an underlying diagnosis of RHD.

We then calculated rates of RHD-attributable morbidity in the general population (figure 2). For the Fiji population aged 0–69 years, we estimated that for every 100 000 person-years there were 14.7 (95% CI 13.6 to 16.0) excess hospitalisations for heart failure, 11.4 for atrial fibrillation (95% CI 10.4 to 12.5), 2.2 for ischaemic stroke (95% CI 1.8 to 2.7) and 1.7 for infective endocarditis (95% CI 1.3 to 2.1) each year (online supplemental table S6). While absolute numbers of RHD-attributable complications peaked in the third decade, the incidence of all but infective endocarditis increased with age. In a multivariate analysis, incidence rates of complications were lower outside the Fijian of Indian Descent population (online supplemental table S7), with this difference being most marked for stroke (incidence rate ratio (IRR) 1.8, 95% CI 1.1 to 2.8, p=0.013) and infective endocarditis (IRR 2.1, 95% CI 1.2 to 3.8, p=0.012). However, contrary to the within cohort risk calculations, overall incidence of complications was higher in women than in men (IRR 1.4, 95% CI 1.3 to 1.6, p<0.001), an effect

that appeared driven predominantly by higher rates of atrial fibrillation (IRR 1.5, 95% CI 1.3 to 1.8, p<0.001).

We estimated a yearly rate of 18.2 RHD-attributable YLDs (95% CI 16.9 to 19.5) per 100 000 person-years in the Fiji population aged 0–69 years (online supplemental table S8). Rates of RHD-attributable disability increased during adolescence and thereafter increased only marginally (online supplemental figure S3). Heart failure alone accounted for 18.7% of YLDs with 28.6% due to other complications while the remaining 52.7% were attributable to RHD without complications. Combined with our previous analyses of RHD-attributable mortality for the same population,[14] we calculated 467.1 DALYs (95% CI 460.5 to 473.7) per 100 000 person-years were lost to RHD in the Fiji population aged 0–69 years during the study period.

### Relationship between RHD-attributable morbidity and mortality

There were 1300 patients in the hospital cohort (a subset of the national cohort) with at least one admission to one of the two regional or one national referral hospitals or outpatient attendance at the national referral hospital during follow-up (table 1). This cohort accounted for 3118 person-years with a median follow-up time of 2.4 years and 238 deaths (18.5%), equating to an overall yearly death rate of 7.2% (95% CI 6.3 to 8.2).

Median survival time was markedly shortened after admission with a complication to 3.6 years after any complication, 2.1 years after heart failure and 2.4 years after stroke (figure 3). Adjusted for age, sex, ethnicity, comorbidity and rural residence, admission for any complication was associated with a fivefold increased risk of death for the remainder of follow-up (HR 5.4, 95% CI 3.4 to 8.8, p<0.001). Moreover, admission for heart failure, atrial fibrillation, stroke and infective endocarditis all had independent effects on risk of death, although the independent effect of atrial fibrillation was only apparent before the onset of heart failure (online supplemental table S9). However, while heart failure and atrial fibrillation had a lasting effect on risk of death, that associated with stroke and infective endocarditis was short-lived and only apparent within the first 28 days of admission.

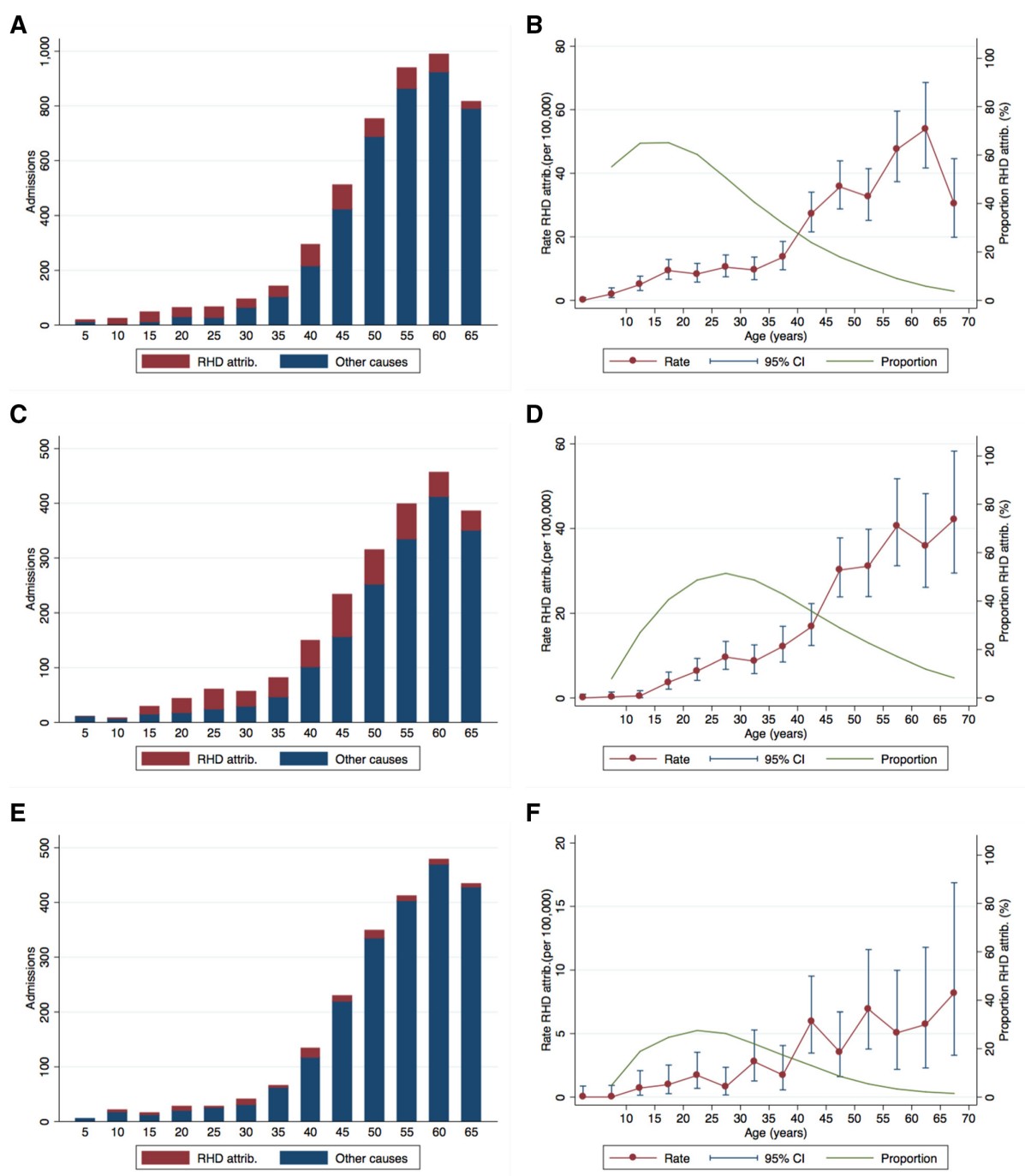

**Figure 2** Hospitalisation for heart failure, atrial fibrillation and stroke in the general population by age. The count of hospitalisations for each complication are shown on the left with the fraction attributable to rheumatic heart disease (RHD) coloured red for: (A) Heart failure, (C) Atrial fibrillation, (E) Ischaemic stroke. Population-based rates of RHD-attributable hospitalisation for each complication are shown on the right with 95% CIs against the left hand axis and the proportion attributable to RHD as a percentage against the right hand axis for: (B) Heart failure, (D) Atrial fibrillation, (F) Ischaemic stroke.

## RHD-attributable morbidity and mortality during maternity

There were 210 women in the maternity cohort with at least one diagnostic code indicating pregnancy or childbirth during follow-up (17.3% of 1215 women in the entire cohort), 204 women with a single pregnancy and 6 women with two pregnancies. A total of 164 deliveries (75.9%) were managed at the national referral hospital, 46 deliveries (21.3%) at two regional referral hospitals and the remaining 6 deliveries at subdivisional or unknown locations. The median age at delivery was 27.0 years (IQR 22.8–31.9).

A total of 123 deliveries (56.9%) had diagnostic codes indicating a complication of labour or delivery of which the most frequent were perineal laceration (74 deliveries), fetal distress (23 deliveries) and umbilical cord complications (10 deliveries). The mode of delivery was indicated in only 50 records (6 indicating caesarean section, 3 forceps or vacuum extraction and

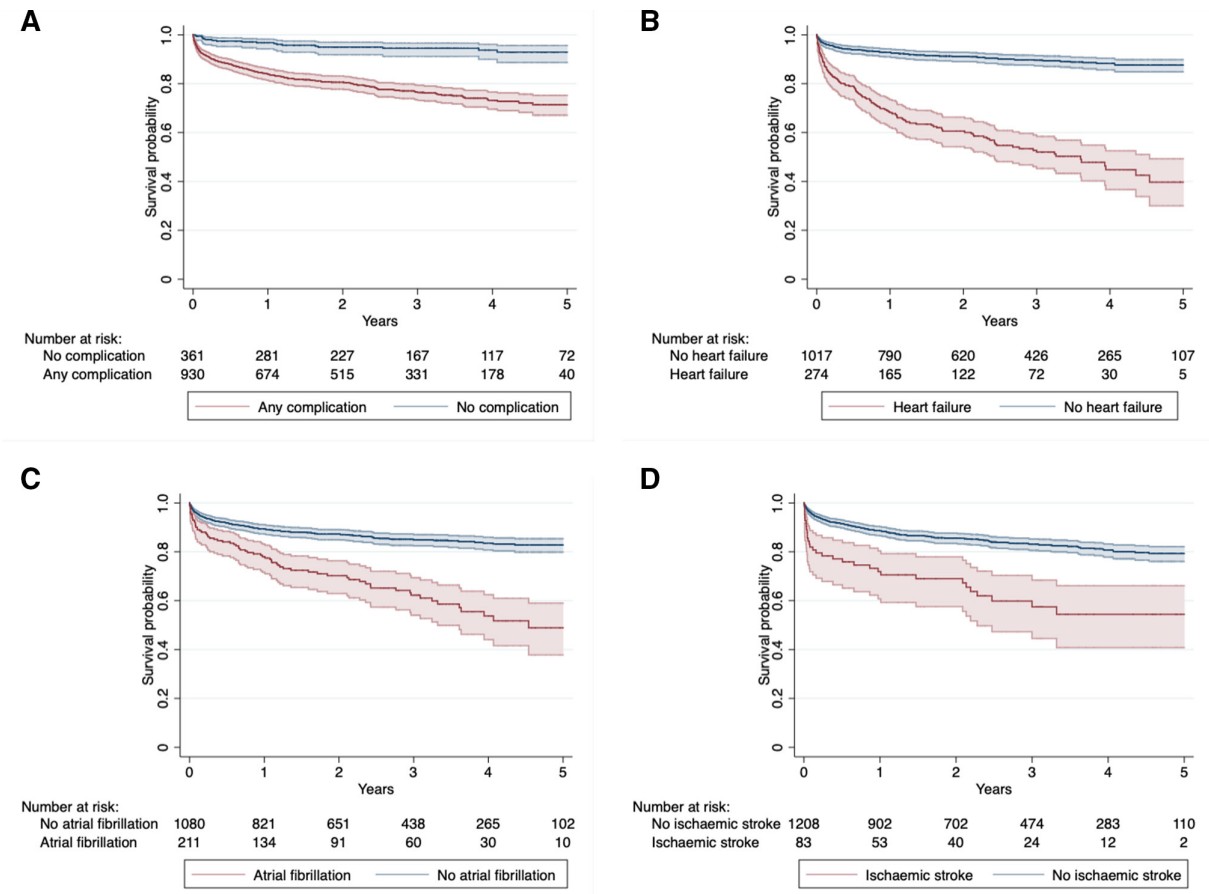

**Figure 3** Survival probability following hospitalisation in the hospital cohort for: (A) Any complication, (B) Heart failure, (C) Atrial fibrillation, (D) Ischaemic stroke. For each event, survival probability is plotted with 95% confidence intervals before and after hospitalisation with the number at risk at each time point in the table.

the remainder spontaneous). Ten of 210 women (4.8%) had a diagnostic code suggesting a prior or coexistent complication of RHD including three each with heart failure, ischaemic stroke and infective endocarditis and two with atrial fibrillation (none of these women died during follow-up). However, overall, 11 of 210 women (5.2%) died during follow-up at a median age 32.2 years, including five after the onset of at least one RHD complication. While none of these deaths occurred within 42 days of delivery, three occurred within 1 year of delivery (ie, late maternal deaths) of which one death was attributable to RHD and two to other causes.

## DISCUSSION

In this nationwide, population-based study, we provide evidence that RHD underlies a substantial proportion of cardiovascular morbidity among children and young adults in Fiji, likely reflecting the situation in LMICs worldwide.

Our analyses provide a considerable amount of new information about the epidemiology of non-fatal RHD complications in LMICs. First, we observed a high burden of RHD-attributable disability among adolescents and young adults, especially women of childbearing age, with

women in the general population at greater risk than men overall. Second, we found a large burden of disability in the population attributable not only to heart failure but also to other complications of RHD—together these other complications accounted for a greater number of YLDs than heart failure alone, which is striking given the reliance on the latter in global summary estimates.[7 9] Third, we found that all four of the non-fatal complications that we assessed were independently associated with an increased risk of death. Consequently, notwithstanding the substantial rates of non-fatal RHD complications in this setting, our study indicates that the dominant public health impact of RHD in this setting is premature death rather than chronic disability.

There are few previously published studies available for comparison to our population-based estimates. Nonetheless, comparison of estimates from our hospital cohort with those from previous hospital-based studies helps benchmark our findings. Focusing on the REMEDY study[15] there is a clear resemblance between the rates of new events in our hospital cohort during the initial 2 years of follow-up and those reported by the REMEDY investigators (online supplemental figure S4).[15] This is illustrated, for example, by the 14.3% of hospital cohort

patients who died within 2 years compared with 12.5% of patients in REMEDY for upper middle-income countries. This comparability underscores the vital importance of our ascertainment of patients away from the hospital setting and increases the generalisability of our findings beyond the Pacific region.

An additional comparison is with the recently reported End Rheumatic Heart Disease in Australia Study of Epidemiology (ERASE) project from Australia, which also identified cases using record-linkage.[18] The most relevant comparison is the yearly cumulative incidence rate of heart failure within a year of diagnosis, which at 3.2% in the Australian study[18] is similar to the 2.5% estimated from our study with overlapping CIs. Additionally, the rates of non-fatal complications that we report are similar to those from a hospital-based cohort reported from the high income Pacific nation of New Caledonia.[31] Beyond this, we and others have previously highlighted the burden of RHD complications in hospital-based and community-based cohorts in the Pacific and beyond.[17 19 32–34] While broadly consistent with our findings, none of these prior studies provided the population-based estimates of disease burden needed to better define the scale of the RHD problem in LMICs.

This analysis extends our previous investigation of cardiac complications of childbirth in Fiji at the national referral hospital.[35] However, the number of complications detected here are much lower than ascertained by case record review. For example, prior or coexistent heart failure was noted in only three women (2%) in the current study compared with 23% in the case note review.[35] While plausible rates are lower in the general population compared with the national referral hospital, it is also likely these lower rates reflect variable recording of non-obstetrical diagnostic codes in the hospital patient information system in this group. Accordingly, the rates of complications that we report here are low relative to previous studies[36–38] and should be interpreted with caution. Nonetheless, the substantial number of deaths in this group during the remainder of follow-up is a stark reminder of the need for new approaches to tackle the burden of RHD among women of childbearing age.[39]

Finally, the GBD study provides the key comparison for our population-based estimates, including those specific to Fiji for the same period.[7 9 12] Our point estimate of YLDs at 18.2 per 100 000 in the population aged 0–69 years is 40% lower than that made by GBD, which, inferred from prevalence, pertains only to heart failure. Importantly, we found a markedly different age-distribution of YLDs to that suggested by GBD, with relatively fewer RHD-attributable YLDs in young adults, likely reflecting markedly shortened survival times after the onset of complications. Accordingly, 96% of DALYs in our study were the result of YLLs, reflecting RHD-attributable deaths among adolescents and young adults,[14] and our DALYs estimate at 467.1 per 100 000 in the population aged 0–69 years is 28% higher than that from GBD. Moreover, since this may reflect the situation in LMICs more widely, it is our view that these discrepancies mandate further work to measure RHD-attributable death and disability in other settings to inform future GBD estimates.

Although our results appear reasonable, they have some limitations. First, notwithstanding the value of linked routine data for epidemiological analyses, there is likely to be some inaccuracy in our data set, including in the precision of coding of discharge diagnoses. Nonetheless, since there is little reason to suspect diagnostic codes would have been used differently in patients with and without RHD, we expect this limitation to have limited impact on our population-based estimates of RHD-attributable morbidity. Future studies might increase accuracy by using algorithms developed to improve ascertainment of RHD cases from routine data, although these algorithms have yet to be validated in data from LMICs.[40]

Second, our data did not distinguish new from pre-existing complications and complications were frequently present at or shortly after diagnosis. To counter this, we limited estimates of the incidence of heart failure (as well as 'any complication' and atrial fibrillation) to the fourth and fifth years of the study on the presumption that almost all of those who developed these events prior to this period would either have either died or been detected during the first 3 years of follow-up. This 'clearance' period is shorter than the 8 years in a recent Australian study[18] but supported by event rates in our cohort (online supplemental figure S2). In addition, to calculate population-based rates, we made comparisons with hospitalisation data from the general population from the same period, which we expect to comprise a similar make up of new and pre-existing complications.

Third, our data lack important information of clinical relevance, including recurrence of acute rheumatic fever, detailed echocardiographic correlates of risk and the impact of therapeutics such as penicillin prophylaxis and anticoagulation. Finally, reflecting external factors, we recognise that several years have now passed since the period from which our multiple sources of routine data were available. Nonetheless, we maintain they provide invaluable insight into the burden of morbidity and mortality attributable to RHD in low-resource settings, not least because it is likely to be a number of years before interventions targeting reduction of childhood *S. pyogenes* infections and improved provision of penicillin prophylaxis impact the burden of disease in the adult population in Fiji.

## CONCLUSIONS

By summarising the burden of non-fatal complications of RHD in Fiji, our study underscores the assertion that effective control of RHD in LMICs could have a substantial impact on the global burden of cardiovascular morbidity and mortality in the young. However, without significant investment in efforts to improve knowledge of RHD epidemiology in LMICs worldwide, targeting and evaluating the impact of control measures will remain

problematic, including, ultimately, delivery of vaccines against the causative *S. pyogenes*.

**Author affiliations**
[1]Department of Infectious Disease, Imperial College London, London, UK
[2]Wellcome Centre for Human Genetics, University of Oxford, Oxford, UK
[3]Department of Obstetrics and Gynaecology, Fiji National University College of Medicine Nursing and Health Sciences, Suva, Rewa, Fiji
[4]Department of Internal Medicine, Fiji National University College of Medicine Nursing and Health Sciences, Suva, Rewa, Fiji
[5]Fiji Ministry of Health and Medical Services, Suva, Rewa, Fiji
[6]Tropical Diseases, Murdoch Children's Research Institute, Parkville, Victoria, Australia
[7]National Centre for Epidemiology and Population Health, Australian National University, Canberra, Australian Capital Territory, Australia
[8]Wesfarmers Centre for Vaccines and Infectious Diseases, Telethon Kids Institute, Perth, Western Australia, Australia

**Acknowledgements** We thank the Ministry of Health and Medical Services and the National Disease Control Programme in Fiji for providing the data sets as well as their employees who compiled them, particularly those at the Ministry of Health Information Unit. We thank and recognise the contribution of the late Dr Isimeli Tukana, formerly Head of Wellness and National Adviser for Non-Communicable Disease at the Fiji Ministry of Health and 21 Medical Services, for his vital contribution to conception and design of the study. We also thank Dr Anne E Hannay (née Miller), Mr Brenton Ward, Dr Rachel Heenan, Ms Tuliana Cua and Ms Maureen Ah Kee for their contribution to the data collection.

**Contributors** TP is the guarantor and accepts full responsibility for the work. TP, JK, MLP, JJF, SMC and ACS conceived the study. TP, JK, LN, SMC and ACS acquired and analysed the underlying data. TP, JK, LN, MLP, KS, JJF, DE, SMC and ACS interpreted the findings. TP wrote the first draft of the manuscript. All authors contributed to and critically revised the manuscript. All authors approved the final version.

**Funding** This work was supported by the Medical Research Council UK (G1100449 to TP), the British Medical Association Foundation for Medical Research (Research Grant 2012 to TP), the Sir Halley Stewart Trust (Research Grant 2013 to TP), the National Institute for Health and Care Research UK (CL-2020-21-001 to TP) and the Wellcome Trust (222098/Z/20/Z to TP). These funders had no role in study design, data collection and analysis, decision to publish or preparation of the manuscript.

**Competing interests** None declared.

**Patient and public involvement** Patients and/or the public were not involved in the design, or conduct, or reporting, or dissemination plans of this research.

**Patient consent for publication** Not applicable.

**Ethics approval** The study was approved by the Fiji National Research Ethics Review Committee (2013-89) and the Oxford University Tropical Research Ethics Committee (1055-13). This study was based on use of routine clinical and administrative data.

**Provenance and peer review** Not commissioned; externally peer reviewed.

**Data availability statement** Data may be obtained from a third party and are not publicly available. The authors did not generate the primary dataset which is the property of the Fiji Ministry of Health and Medical Services. Researchers wishing to access Fiji Ministry of Health data should send a request in writing to the Permanent Secretary for Health, Fiji Ministry of Health and Medical Services, Dinem House, 88 Amy Street, PO Box 2223, Government Buildings, Suva, Fiji Islands (see also: http://www. health.gov.fj). Use of Fiji Ministry of Health data would also be subject to approval by the Fiji National Health Research Committee.

**ORCID iDs**
Tom Parks http://orcid.org/0000-0002-1163-8654
Mai Ling Perman http://orcid.org/0000-0002-3511-3297
James J Fong http://orcid.org/0000-0002-3962-7105
Daniel Engelman http://orcid.org/0000-0002-4909-1287
Samantha M Colquhoun http://orcid.org/0000-0002-6750-1147
Joseph Kado http://orcid.org/0000-0003-3121-4397

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
