## [Reviewer comments · BMJ Open]

ARTICLE DETAILS

TITLE (PROVISIONAL)	Population-Based Assessment of Cardiovascular Complications of Rheumatic Heart Disease in Fiji: a Record-linkage Analysis
AUTHORS	Parks, Tom; Narube, Litia; Perman, Mai Ling; Sakumeni, Kelera; Fong, James; Engelman, Daniel; Colquhoun, Samantha; Steer, Andrew; Kado, Joseph

VERSION 1 – REVIEW

REVIEWER	Ali, Fatima Aga Khan University, Pediatrics and child health
REVIEW RETURNED	23-Jan-2023

GENERAL COMMENTS	The study is interesting and the paper is written comprehensively enough to understand. The objectives and methodology have provided adequately that the report does offer enough details to reproduce the experiments. No major revision from my side. However, there are a few comments 1) It was confusing to understand the National cohort and hospital cohort. Is the hospital cohort being a subset of the national cohort?2) How many hospitals were checked for this population-based study? Did all hospitals from the government and private sectors report in the hospital information system?3) The use of nonfatal term gives the impression that cardiovascular complications like heart failure, atrial fibrillation, etc are minor complications, which in reality are not they are fatal ultimately if not managed. The replacement of nonfatal to cardiovascular complications would represent better, I believe.4) Overall, very nice contribution to the literature on RHD.
---

REVIEWER	Chen, Qing Southern Medical University School of Public Health, Department of Epidemiology
REVIEW RETURNED	24-Jan-2023

GENERAL COMMENTS	This study provided the comprehensive population-based rates of nonfatal complications of rheumatic heart disease in Fiji using a national cohort. The manuscript is well-written and please find my suggestions below for your considerations. 1. In Abstract, Fiji is an upper-middle-income country, but there would be confusion in the conclusion where the authors declared that the burden of RHD-attributable morbidity in Fiji may reflect the situation in low- and middle-income regions. The conclusions of the study should be drawn with caution since, in addition to
---

	economic condition, other factors such as geography, lifestyle, and nutrition may affect the RHD-related morbidity. 2. In Page 10, authors of this study mentioned that the yearly cumulative incidence rate of any complication except infective endocarditis was increasing with age. Chi-square test for trend is strongly recommended to employ.\ 3. Although the authors have clearly described the cumulative incidence rate for both each and any complications of RHD, these rates for co-complications of RHD are particularly noteworthy as they would be linked to RHD-attributable mortality.
--	--

VERSION 1 – AUTHOR RESPONSE

Reviewer: 1

Dr. Fatima Ali, Aga Khan University

Comments to the Author:

Manuscript ID: bmjopen-2022-070629,

Title: "Population-Based Assessment of Nonfatal Complications of Rheumatic Heart Disease in Fiji: A Record-linkage Analysis."

The study is interesting and the paper is written comprehensively enough to understand. The objectives and methodology have provided adequately that the report does offer enough details to reproduce the experiments.

No major revision from my side. However, there are a few comments

1) It was confusing to understand the National cohort and hospital cohort. Is the hospital cohort being a subset of the national cohort?

It is correct that the 'hospital' cohort is a subset of the 'national' cohort as described in the second paragraph of the methods. The relationship of these cohorts is also indicated in Supplementary Figure S1. For greater clarity, we have added the text "a subset of the national cohort" to the opening sentence of Section 4.3.

2) How many hospitals were checked for this population-based study? Did all hospitals from the government and private sectors report in the hospital information system?

The analysis includes all government hospitals in the country including one national and two regional referral hospitals but not the relatively small private sector. For clarity, we have added the following sentence to the second paragraph of Section 3.2: "Data were available from all government funded hospitals, including the two regional and one national referral hospitals, but not from the smaller private sector."

3) The use of nonfatal term gives the impression that cardiovascular complications like heart failure, atrial fibrillation, etc are minor complications, which in reality are not they are fatal ultimately if not managed. The replacement of nonfatal to cardiovascular complications would represent better, I believe.

The term “nonfatal” is used for consistency with the Global Burden of Disease study, which assesses morbidity due to events such as heart failure and atrial fibrillation under this umbrella term. We have removed the word nonfatal from the title but otherwise for this reason would prefer to keep it in place elsewhere in the manuscript.

4) Overall, very nice contribution to the literature on RHD.

Reviewer: 2

Dr. Qing Chen, Southern Medical University School of Public Health

Comments to the Author:

This study provided the comprehensive population-based rates of nonfatal complications of rheumatic heart disease in Fiji using a national cohort. The manuscript is well-written and please find my suggestions below for your considerations.

1. In Abstract, Fiji is an upper-middle-income country, but there would be confusion in conclusion where the authors declared that the burden of RHD-attributable morbidity in Fiji may reflect the situation in low- and middle-income regions. The conclusions of the study should be drawn with caution since, in addition to economic condition, other factors such as geography, lifestyle, and nutrition may affect the RHD-related morbidity.

Fiji is an upper-middle-income country and so within the wider bracket of low- and middle-income countries. As outlined in the Discussion (paragraph 3) and indicated in Supplementary Figure S4, there is considerable concordance between the rates we report from Fiji and those reported by the REMEDY investigators from sites across a range of low- and middle-income countries. Thus, while we agree that numerous factors affect RHD-related morbidity, we maintain that our estimates from Fiji are likely relevant to RHD endemic settings outside the Pacific region. Nonetheless, to give a more cautious interpretation, we have replaced the word “likely” with “potentially” in the Conclusions of the Abstract.

2. In Page 10, authors of this study mentioned that the yearly cumulative incidence rate of any complication except infective endocarditis was increasing with age. Chi-square test for trend is strongly recommended to employ.

The hazard associated with increasing age has been calculated in a multivariate competing-risks regression model for each complication as detailed in Supplementary Table S5.

3. Although the authors have clearly described the cumulative incidence rate for both each and any complications of RHD, these rates for co-complications of RHD are particularly noteworthy as they would be linked to RHD-attributable mortality.

We have added additional information related to co-complications to Section 4.1: “Furthermore, 220 (10.4%) patients had admissions for more than one complication of which heart failure and atrial fibrillation in 154 (7.3%) followed by ischaemic stroke and atrial fibrillation in 46 (2.2%) were the most frequent combinations.” Additionally, both the DALYs calculations described in Section 4.2 and the survival analyses described in Section 4.3 take co-complications into consideration. The latter

specifically investigates interactions showing that all four of the complications considered in our analyses have independent effects on risk of death but with an interaction between heart failure and atrial fibrillation.

VERSION 2 – REVIEW

REVIEWER	Ali, Fatima Aga Khan University, Pediatrics and child health
REVIEW RETURNED	03-Feb-2023
GENERAL COMMENTS	In a good status

VERSION 2 – AUTHOR RESPONSE